# Systematic review of endovascular stent grafting versus open surgical repair for the elective treatment of arch/descending thoracic aortic aneurysms

Andrew McCarthy ,[1] Joanne Gray,[1] Priya Sastry,[2] Linda Sharples,[3] Luke Vale,[4] Andrew Cook ,[5,6,7] Peter Mcmeekin,[1] Carol Freeman ,[8] Pedro Catarino,[9] Stephen Large[9]

For numbered affiliations see end of article.

**Correspondence to**
Andrew McCarthy;
andrew2.mccarthy@
northumbria.ac.uk

## ABSTRACT

**Objective** To review comparisons of the effectiveness of endovascular stent grafting (ESG) against open surgical repair (OSR) for treatment of chronic arch or descending thoracic aortic aneurysms (TAA).

**Design** Systematic review and meta-analysis

**Data sources** MEDLINE, EMBASE, CENTRAL, WHO International Clinical Trials Routine data collection, current controlled trials, clinical trials and the NIHR portfolio were searched from January 1994 to March 2020.

**Eligibility criteria for selective studies** All identified studies that compared ESG and OSR, including randomised controlled trials (RCTs), quasi-randomised and non-RCTs, comparative cohort studies and case–control studies matched on main outcomes were sought. Participants had to receive elective treatments for arch/descending (TAA). Studies were excluded where other thoracic aortic conditions (eg, rupture or dissection) were reported, unless results for patients receiving elective treatment for arch/descending TAA reported separately.

**Data extraction and synthesis** Data were extracted by one reviewer and checked by another. Risk of Bias was assessed using the ROBINS-I tool. Meta-analysis was conducted using random effects. Where meta-analysis not appropriate, results were reported narratively.

**Results** Five comparative cohort studies met inclusion criteria, reporting 3955 ESG and 21197 OSR patients. Meta-analysis of unadjusted short-term (30 day) all-cause mortality favoured ESG (OR 0.75; 95% CI 0.55 to 1.03)). Heterogeneity identified between larger and smaller studies. Sensitivity analysis of four studies including only descending TAA showed no statistical significance (OR 0.73, 95% CI 0.45 to 1.18)), moderate heterogeneity. Meta-analysis of adjusted short-term all-cause mortality favoured ESG (OR 0.71, 95% CI 0.51 to 0.98)), no heterogeneity. Longer-term (beyond 30 days) survival from all-cause mortality favoured OSR in larger studies and ESG in smaller studies. Freedom from reintervention in the longer-term favoured OSR. Studies reporting short-term non-fatal complications suggest fewer events following ESG.

**Conclusions** There is limited and increasingly dated evidence on the comparison of ESG and OSR for treatment of arch/descending TAA.

**PROSPERO registration number** CRD42017054565.

## Strengths and limitations of this study

► This is the first systematic review to investigate two-armed studies looking only at elective endovascular stent grafting versus open surgical repair for treatment of chronic arch or descending thoracic aortic aneurysms. Other reviews include dissections, ruptures and treatment for other conditions.

► This systematic review inclusion criteria were broad and sought to include evidence from both randomised and non-randomised study designs.

► Statistical analysis of non-fatal complications was not possible due to the limited and inconsistent reporting of such complications.

## INTRODUCTION
### Rationale

A thoracic aortic aneurysm (TAA) is an abnormal expansion of the aorta at any part of its course from the heart to the diaphragm. Aneurysms manifest with acute symptoms or they may be chronic. If untreated, TAA can continue to expand and the annual risk of aneurysm rupture increases from 2% in aneurysms smaller than 5 cm in diameter to 7% in aneurysms over 6 cm in diameter.[1] The majority of aneurysms (and therefore studies) centre on the ascending or abdominal segments of the aorta, arch and descending aneurysms are relatively understudied. Therefore, this review focused on chronic aneurysms affecting the arch or descending thoracic aorta (DTA).

There are currently two main methods of repair for arch/descending TAA: open surgical repair (OSR) and endovascular stent grafting (ESG). OSR involves replacing the aneurysmal segment of aorta with a surgical graft during open surgery using heart-lung bypass. As the arch and descending aorta supply blood to the

brain and spinal cord, OSR is high risk, with reported mortality and paraplegia rates of approximately 5% and 10%,[2] respectively. In ESG, a stent is placed across the aneurysm, providing a new conduit through which the blood flows. ESG is most often selected for patients who are considered at high risk of death or permanent disability from OSR. However, some patients may not have a suitable anatomy/morphology for ESG procedures.[2] Concerns have also been raised about long-term durability and safety of the implanted devices[3 4] due to complications such as stent migration or endoleaks, with the incidence of late endoleaks (leaks around or through the stent graft that occur more than 1 year after insertion) reported as being between 2.5% and 27.8%.[5] The need for reintervention due to technical failures of the stent following ESG is also high (23% at 5 years).[6 7] With each reintervention there is an added risk of complication either due to the increased complexity of the procedure or deteriorating health of the patient.[2]

One previous systematic review comparing only elective OSR and ESG for descending and arch TAAs has been reported.[8] This systematic review[8] was restricted to data from randomised controlled trials (RCTs) in which patients were randomly assigned to ESG and OSR, but no eligible trials were identified when searching relevant data based up to January 2016. There are other systematic reviews[9 10] that have included non-randomised studies; however, these reviews also included emergency cases and other thoracic aortic pathologies (eg, dissection or rupture). As such, they may not represent a true comparison of elective ESG and OSR procedures for DTA aneurysms. Given the lack of RCT evidence, this systematic review includes non-randomised comparative studies of the relative effectiveness of OSR or ESG.

The decision to operate requires a favourable balance between the risks and benefits of that intervention compared with those attached to the expected outcomes of the disease. This balance is in turn influenced by a number of factors. Aneurysm size correlates to the risk of death (if left untreated) and so patients are usually referred once the aneurysm crosses a chosen threshold. At that point, patient fitness (and life expectancy), the reality or possibility of an intrinsic weakness affecting the whole aorta, the technical feasibility of the proposed intervention and patient wishes, come into play. This systematic review takes a step back from these issues and examines only the outcomes for patients in whom a decision to intervene has been made. This review is an essential starting point for the analysis of effective treatments for thoracic aortic aneurysms.

## Objectives

The aim of this systematic review was to assess the effectiveness of ESG compared with OSR as a treatment for chronic arch or descending TAA.

## METHODS

### Protocol and registration

The review protocol was registered in the PROSPERO database (CRD42017054565).[11]

### Patient and public involvement

There was no patient and public involvement in the whole process of conducting this review.

### Eligibility criteria

#### Study designs

All identified studies that directly compared ESG and OSR, including RCTs, quasi-randomised and non-RCTs, comparative cohort studies and case–control studies matched on main outcomes were sought.[12]

Non-randomised studies that controlled for selection bias were included—irrespective of study size. Acceptable methods for addressing selection bias included matching patients at baseline for important covariates, matching patients using propensity scores or adjustment for important covariates during analysis. Non-randomised studies that did not use these methods but included larger sample sizes (more than 250 patients) were also included as they have the potential to reduce sampling variation relative to smaller unmatched studies.[13]

Non-English language papers were noted and, where possible, data were extracted from the abstract. No translation of full-text articles was possible.

### Study participants

Study participants had to receive elective treatments for arch and/or DTA aneurysms. Where emergency interventions or other thoracic aortic conditions (such as intramural haematoma, rupture and dissection) were reported, studies were excluded, unless results for patients receiving elective treatment for arch/descending TAA were reported separately. Studies that reported secondary (reintervention) procedures for arch/descending TAAs were included only if the original, primary procedure for each participant could be identified.

### Study interventions

OSR was defined as replacement of the aneurysmal aorta with a prosthetic conduit via a sternotomy or thoracotomy with circulatory support. ESG was defined as transluminal introduction of a stent-graft under X-ray guidance.

### Outcome measures

This review assessed short-term and long-term clinical effectiveness. Short-term was defined as either within 30 days of the intervention and/or time until discharge following the intervention. Since most studies did not report long-term outcomes conditional on surviving the short-term period, long-term analyses covered the time from intervention until the end of follow-up. Primary outcomes were all-cause mortality and aneurysm-related mortality. Secondary outcomes were reinterventions and non-fatal complications. The latter were subclassified as

vascular, neurological, cardiac, respiratory and 'other' complications.

## Information sources

MEDLINE, EMBASE, CENTRAL, WHO International Clinical Trials Routine data collection, Current Controlled Trials, Clinical Trials, and the UK National Institute for Health Research portfolio were searched. The search strategy (online supplemental appendix 1) was adapted as appropriate for each database (for example to allow for variations in controlled vocabulary terms and syntax.[14] Electronic searches identified publications between 1994, date of first reported case of ESG for descending and arch TAA,[15] and March 2020.

Websites of regulatory bodies and Health Technology Assessment agencies were screened for unpublished relevant reports. A citation search of all included studies was conducted via Web of Science. Studies from systematic reviews that satisfied the inclusion criteria of this review were screened for eligibility.

## Study selection

All titles and abstracts were screened by one reviewer (AM). A second investigator (JG) reviewed 10% of all titles and abstracts classed as eligible or not eligible and all titles and abstracts classified as unclear by the first reviewer. The full text of selected studies was assessed by one investigator (AM) and classified as relevant, not relevant or unclear according to the inclusion criteria. A second investigator (JG) reviewed 10% of all papers, as well as all studies deemed unclear. Disagreements at either stage were resolved by discussion and where necessary a third reviewer (PM) arbitrated. Where required, a clinical expert (PS) was asked to screen papers.

## Data collection process

A data extraction form was developed based on The Cochrane Collaboration data collection form.[16] Data were extracted by one reviewer (AM) and checked by a second reviewer (JG) for accuracy. Disagreements were resolved by discussion or arbitration by a third reviewer (PM).

## Data items

Reported short-term and longer-term all-cause mortality, as well as short-term and longer-term aneurysm-related mortality were collected. All reported complications were recorded and grouped according to expert clinical guidance as described above as neurological, cardiac, respiratory or 'other' complications, with endoleaks following ESG recorded separately.

## Risk of bias in individual studies

Risk of bias tools used were specific to the study design. As described later only non-randomised study designs were identified. Therefore, the ROBINS-I tool[17] ("Risk Of Bias In Non-randomised Studies - of Interventions") was used. The ROBINS-I tool classifies the risk of bias in seven domains (confounding; selection of participants; classification of interventions; deviations from intended interventions; missing data; measurement of outcomes; selection of the reported results) and overall as either critical, severe, moderate, low or unclear.

## Synthesis of results

Point estimates and precision of the incidence of short-term and long-term events and unadjusted and adjusted comparisons (ORs and HRs) were recorded. Meta-analysis of short-term event comparisons was based on the OR with 95% CI and conducted using RevMan.[18] An overall treatment effect was calculated for the log (OR) of each outcome using a (normal) Mantel-Haenszel (M-H) random-effects model. Where comparisons could not be made, data were tabulated and reported narratively.

Heterogeneity between studies was assessed according to Cochrane guidelines, on clinical criteria, by visual inspection of plots of the data, from the $\chi^2$ test for heterogeneity, and the I-squared statistic. An $I^2$ statistic of 25%–50% indicated low heterogeneity; 50%–75% indicated moderate heterogeneity and over 75% indicating significant heterogeneity.

## RESULTS

## Study selection

The search identified 5163 unique studies. Of these, 5078 studies were not eligible. Ineligibility was primarily because of irrelevant anatomy (ascending, abdominal or peripheral aneurysms), irrelevant pathology (acute dissection or other acute aortic syndromes) or because the study pooled data from a mixture of anatomical and pathological situations such that data on chronic arch and DTA aneurysms could not be extracted with confidence. Thus, 85 papers underwent full-text screening. Seventy-five papers were rejected at this stage leaving 10 papers to be included in the full review. Six papers reported data from the Gore TAG Trial,[19–24] leaving five unique comparative studies that fulfilled the inclusion criteria. Further details of the selection process are shown in the Preferred Reporting Items for Systematic Reviews and Meta-Analyses diagram (figure 1).

One study, Goodney et al[25] reported both an unadjusted analysis for the unadjusted cohort and an analysis of a subgroup of propensity score matched patients. Data from the unadjusted cohort were used in all meta-analyses except the adjusted mortality meta-analysis, which used the propensity matched data. Brief summaries of the identified studies are given in table 1, with detail provided in online supplemental appendix 2. Overall, the five studies included data on 3955 ESG and 21 197 OSR patients and all studies were conducted in Europe or the USA. Two studies had non-overlapping or only partially overlapping recruitment periods between the ESG and OSR intervention groups (Gore Tag[19–24] and Piffaretti et al).[26] There were clear differences in patient characteristics between the two treatment groups. ESG patients were consistently older, with average age differences ranging

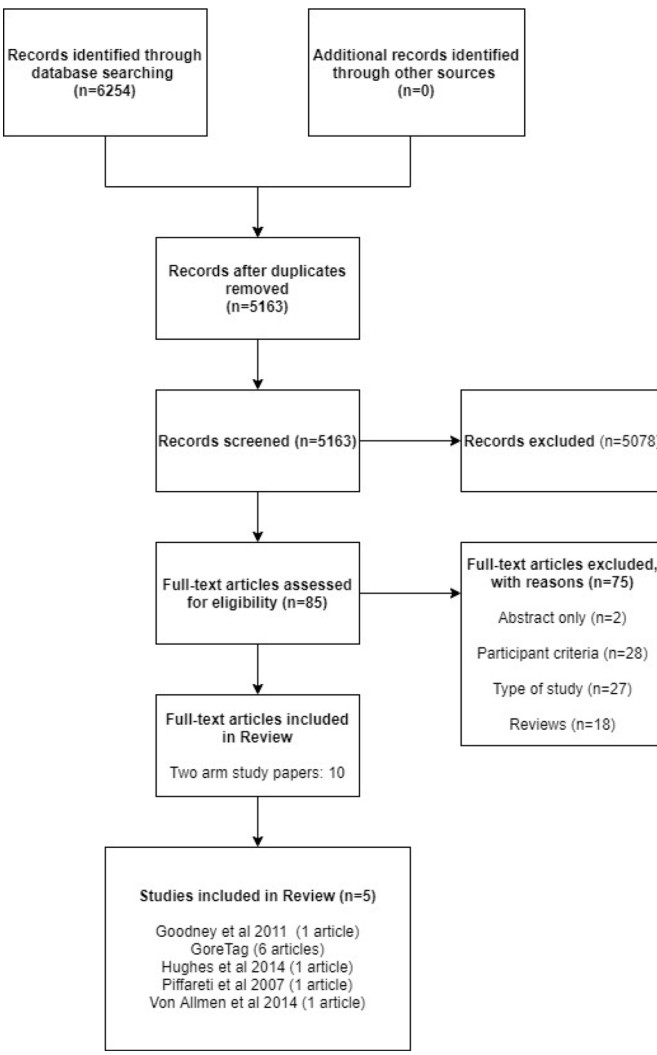

**Figure 1** PRISMA diagram. PRISMA, Preferred Reporting Items for Systematic Reviews and Meta-Analyses

from 2 to 9 years in all but one analysis (Goodney *et al*[25] the propensity matched cohort). Similarly, in most studies the proportion of men was greater in the ESG group.

### Risk of bias within studies

The propensity-matched analysis of Goodney *et al*[25] was judged to be of overall moderate risk of bias and four studies were at severe risk of bias due to the potential for confounding (online supplemental appendix 3).

Two large cohorts derived from UK and USA administrative databases (von Allmen *et al*[27] and Hughes *et al*[28]) did not attempt to address selection bias at the design stage, but did adjust for some confounders in the analysis. A third study (Goodney *et al*,[25] using USA administrative data, included both unadjusted population and analysis of a subset of the data defined by propensity score matching. Propensity score methods rely on all important confounders being available and included.[29] As this study did not include all confounders considered important, such as aneurysm size, this study was judged to have moderate risk of bias due to confounding.

One study (Gore TAG)[19–24] included only ESG patients who would also have been eligible for OSR and the another study (Piffaretti *et al*)[26] included only OSR patients who would be suitable for ESG, both according to clinician judgement. Since suitability was decided retrospectively, such matching may not be robust.

A further potential bias with each study was selective reporting. Although the measurements and analyses of the outcomes were clearly defined within each study, no evidence of a prestudy analysis plan was identified.

### Synthesis of results
#### Short-term death from any cause (30-day or in hospital)
*Unadjusted mortality*
All five studies[19–28] reported unadjusted 30 days or in-hospital mortality (figure 2). Overall, the odds of short-term mortality were lower in the ESG group than the OSR group (OR: 0.75; 95% CI 0.55 to 1.03). Moderate heterogeneity was present in the overall analysis ($I^2$=40%), mainly due to differences between the two smaller studies[19–24 26] and the three larger studies,[25 27 28] with a large $I^2$ for subgroup differences between the larger studies subgroup and the smaller studies subgroup of 84.5%.

For the three larger studies[25 27 28] (3798 ESG and 21 092 OSR patients), the treatment effect for death was smaller (OR 0.84 95% CI 0.72 to 0.98)) than that for the two smaller studies[19–24 26] (157 ESG and 105 OSR patients, combined OR 0.18 95% CI 0.06 to 0.59)). There was no evidence of heterogeneity in results between these subgroups ($I^2$=0).

*Sensitivity analysis*
Hughes *et al*[28] was the only study to include aortic arch aneurysms. Arch aneurysms tend to be more complex procedures and are mostly conducted using OSR. A sensitivity analysis excluding this study was, therefore, conducted (figure 3).

While the point effect estimates all favoured ESG, the overall pooled OR (OR 0.73; 95% CI 0.45 to 1.18)) did not reach statistical significance (p=0.2). Moderate heterogeneity was present in the overall analysis ($I^2$=42%), mainly due to differences between the two smaller studies and the two larger studies, with a large $I^2$ for subgroup differences between the larger studies subgroup and the smaller studies subgroup of 65.1%.

For the two larger studies[25 27] (3086 ESG and 12 837 OSR patients), the treatment effect for death was smaller (OR 0.85; 95% CI 0.72 to 1.00)) than that for the two smaller studies[19–24 26] (157 ESG and 105 OSR patients, combined OR 0.20; 95% CI 0.04 to 1.06)). There was no evidence of heterogeneity in results within the large subgroup ($I^2$=0) and evidence of low heterogeneity withing the small studies subgroup ($I^2$=26%).

*Adjusted mortality*
The three larger studies[25 27 28] included adjusted analysis of short-term mortality (figure 4). Goodney *et al*[25] used stratified propensity matching. Hughes *et al*[28] adjusted

**Table 1** Study characteristics

| Study | von Allmen et al 2014[27] | Goodney et al 2011[25] (Two analyses*) | Gore TAG | Hughes et al 2014[28] | Piffaretti et al 2007[26] |
|---|---|---|---|---|---|
| **Mthods** | | | | | |
| **Study design** | Comparative cohort | Comparative cohort | Comparative cohort | Comparative cohort | Comparative cohort |
| **Method of matching** | None | 1. None<br>2. Propensity matched | ESG patients judged eligible for OSR | None | OSR patients judged eligible for ESG |
| **Participants** | | | | | |
| **Country** | UK | USA | USA | USA | Italy |
| **Setting** | Multicentre—Hospital episode statistics | Multicentre—Medicare data | Multicentre | Multicentre | Single centre |
| **Time frame** | 2006–2011 | 1. 1998–2007<br>2. 2003–2007 | ESG 1999- 2001, OSR pre 1999 (n=44) and 1999–2001 (n=50) | 1998–2007 | ESG 2000–2007, OSR 1996–2000 |
| **Recruitment** | Total 618<br>ESG 354<br>OSR 264 OSR | 1. Total 13 998 (ESG 2433, OSR 11,565)<br>2. Total 1100 (ESG 550, OSR 550) | Total 234<br>ESG 140<br>OSR 94 | Total 8967<br>ESG 712<br>OSR 8255 | Total 28<br>ESG 17<br>OSR 11 |
| **Age summary** | Median (IQR)<br>ESG 73 (66,78), OSR 71 (63–76) | Mean (95% CI)<br>1. ESG 75.9 (75.6 to 76.1), OSR 73.8 (73.7 to 73.9)<br>2. ESG 71.1 (70.8 to 71.4), OSR 70.7 (70.7 to 71.1) | Mean (SD)<br>ESG 75.9 (10.4), OSR 68.2 (10.2) | Median (IQR)<br>ESG 72 (63, 78), OSR 63 (52, 72) | Mean (SD)<br>ESG 66 (10), OSR 61 (13) |
| **Proportion male** | ESG 232 (65.5%), OSR 137 (51.9%) | % (95% CI):<br>1. ESG 58.7 (56.7 to 60.7), OSR 55.4 (54.8 to 56.3)<br>2. ESG 64.0 (59.9 to 68.0), OSR 68.1% (64.2 to 72.0) | ESG 90 (57%), OSR 48 (51%) | 5415 (61.3%) ESG, 437 (65.6%) OSR | ESG 14 (82%), OSR 8 (73%) |
| **Outcomes** | | | | | |
| **Short term** | Death from any cause Reintervention all-cause mortality (unadjusted | Death from any cause | Death from any cause Complications | Death from any cause Complications | Death from any cause Complications |
| **Long term** | Time to death from any cause Time to aortic-related death Time to aortic-related reintervention | Time to death from any cause | Time to death from any cause Time to aneurysm-related death aneurysm-related mortality Time to reintervention Time to major adverse event | Not reported | Time to death from any cause |

*1. Unmatched cohort, 2. Propensity matched cohort.
ESG, endovascular stent grafting; OSR, open surgical repair.

for preoperative comorbidities, independent predictors of mortality and postoperative complications and the von Allmen et al[27] study adjusted for age and sex. The OR for ESG, relative to OSR, for these three studies combined was 0.71, 95% CI 0.51 to 0.98). No evidence of heterogeneity was identified in this meta-analysis (I$^2$=0%)

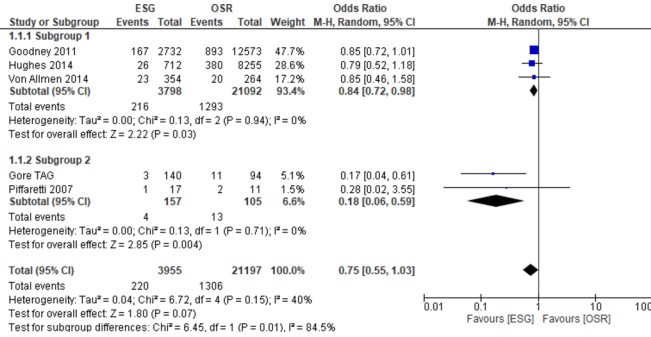

**Figure 2** Forest plot of unadjusted short-term all-cause mortality. ESG, endovascular stent grafting; OSR, open surgical repair.

## All-cause mortality (following hospital discharge or after 30 days)

Meta-analysis of the longer-term all-cause mortality was not possible as follow-up varied between the four studies that reported this outcome (Gore Tag,[19–24] Goodney et al,[25] Piffaretti et al[26] and von Allmen et al.[27] The results in the larger studies (Goodney et al,[25] von Allmen et al[27] differed from those in the smaller studies (Gore Tag[19–24] and Piffaretti et al.[26] We note that the analyses included the short-term deaths as well as those occurring beyond hospital discharge or after 30 days.

von Allmen et al[27] reported time to death from any cause up to 5 years, throughout which OSR had higher overall survival. Adjusting for age and sex, ESG had higher odds of death at 1 year (OR 1.10; 95% CI 0.70 to 1.73); p=0.667). Additionally, the HR for ESG relative to OSR up to 5 years, adjusting for age and sex, was HR=1.45 (95% CI 1.08 to 1.94), p=0.013).

Goodney et al[25] also reported time to death from any cause up to 5 years. Adjusting for age, sex, race, era of procedure and Charlson Comorbidity Score, 5-year survival for OSR in the unmatched cohort was higher than for ESG (89% vs 79%, log-rank p<0.0001). In the propensity-matched cohort, 5 years life-table analysis reported higher survival in the OSR group (81%–95% CI 77% to 85%)) than for the ESG group (73%–95% CI 68% to 76%)) (log-rank p=0.007).

In the Gore TAG study,[19–24] time to death from any cause over 5 years did not differ between the ESG and OSR

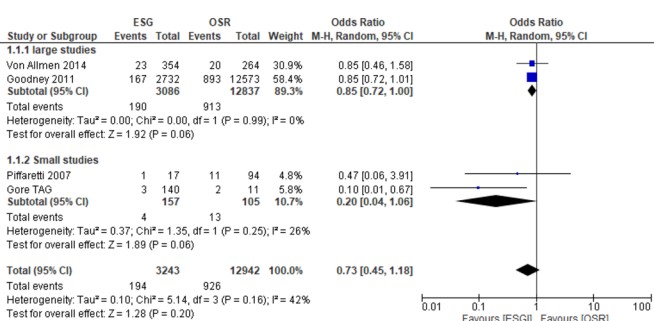

**Figure 3** Forest plot of unadjusted short-term all-cause mortality. sensitivity analysis. ESG, endovascular stent grafting; OSR, open surgical repair

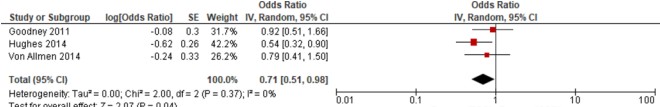

**Figure 4** Forest plot of adjusted short-term all-cause mortality. ESG, endovascular stent grafting; OSR, open surgical repair.

groups, with survival probability at 5 years in both groups of 63% (log rank p=0.625). Piffaretti et al[26] reported survival overall in the ESG group up to 5 years and the OSR group up to 13 years. For the 5 years in which survival is reported in both groups, ESG had a higher survival compared with OSR although this was based on only four deaths, and the follow-up periods differed substantially (log rank p>0.05).

## Aneurysm/aortic-related mortality

Meta-analysis of aneurysm/aortic-related mortality could not be conducted. No studies reported aneurysm-related mortality during hospitalisation or up to 30 days separately to the long-term. Von Allmen et al 2014[27] defined aortic-related mortality as any aortic-related death occurring within 5 years of the index operation according to the International Classification of Diseases (ICD-10) codes referring to aorta, or cause of deaths closely related to complications of surgery (eg, paraplegia). Time to aortic-related death did not differ between OSR group than the ESG group during the 5 years follow-up (log-rank p=0.270). This study[27] also reported adjusted (for age and sex) 5 years HR of 1.27 (95 CI 0.81 to 1.98; p=0.298) for ESG patients relative to OSR patients.

Gore TAG[19–24] defined aneurysm-related mortality as deaths that occurred in hospital or within 30 days of initial procedure or any reintervention or a death due to the aneurysm or the treatment device and reported up to 5 years, with longer time to aneurysm-related death in the ESG group over this period (log-rank p=0.012).

## Short-term reinterventions

One study reported aortic-related re-interventions in 21 (5.9%) ESG patients and 4 (1.5%) OSR patients, an adjusted OR of 2.97 (95% CI 1.09 to 8.10), p=0.033) for ESG versus OSR (von Allmen et al).[27]

## Long-term reinterventions

Two studies reported this outcome (von Allmen et al[27] and Gore TAG.[19–24] von Allmen et al[27] reported time to aortic-related reinterventions up to 5 years. Time to reintervention was longer in the OSR group compared with the ESG group (log-rank p=0.007), with adjusted (for age and sex) HR at 5 years of 1·70 (95% CI 1.11 to 2.60), p=0·014). In Gore TAG,[19–24] time to reintervention over 5 years was longer in the ESG group (log-rank p=0.011). We note that these analyses included short-term reinterventions.

## Complications

Table 2 summarises the short-term complications reported in all five studies.[19 25–28] Most studies reported a small number of neurological, pulmonary, cardiac and

**Table 2** Short-term complications

| Study | Gore TAG | | Piffaretti *et al*[26] | | Hughes *et al*[28] | |
|---|---|---|---|---|---|---|
| Arm (participants) | ESG (n=140) | OSR (n=94) | ESG (n=17) | OSR (n=11) | ESG (n=712) | OSR (n=8255) |
| Neurological | | | | | | |
| Paraplegia | NR | NR | NR | NR | NR | NR |
| Paraplegia/paraperisis | 4 (3%) | 13 (14%) | NR | NR | NR | NR |
| Cerebral vascular accident | 5 (4%) | 5 (4%) | 2 (9%) | 1 (12%) | NR | NR |
| Neurological-unspecified type | NR | NR | NR | NR | 20 (2.8%) | 273 (3.3%) |
| Respiratory | | | | | | |
| Respiratory failure | 5 (4%) | 19 (20%) | NR | NR | NR | NR |
| Pneumonia | NR | NR | 2 (12%) | 3 (27%) | NR | NR |
| Pulmonary-unspecified type | NR | NR | NR | NR | 17 (2.4%) | 462 (5.6%) |
| Cardiac | | | | | | |
| Myocardial Infarction | 0 (0%) | 1 (1%) | 0 (0%) | 1 (9%) | NR | NR |
| Cardiac-unspecified type | NR | NR | NR | NR | 28 (2.9%) | 1252 (15.2%) |
| Endoleaks | | | | | | |
| Endoleaks | 12 (8.5%) | NR | NR | NR | NR | NR |
| Other | | | | | | |
| Peripheral vascular disease | 20 (14%) | 4 (4%) | NR | NR | NR | NR |
| Renal failure | 2 (1%) | 12 (13%) | NR | NR | NR | NR |
| Wound infection/dehiscence | 5 (4%) | 10 (11%) | NR | NR | NR | NR |
| Gastrointestinal complication | 3 (2%) | 6 (6%) | NR | NR | NR | NR |
| Postimplant syndrome | NR | NR | 4 (18%) | 0 (0%) | NR | NR |

Goodney *et al*[25] and von Allmen *et al*[27] did not report short-term complications.
ESG, endovascular stent grafting; NR, not reported; OSR, open surgical repair.

other complications. However, one larger study (Hughes *et al*)[28] reported lower odds for ESG patients for neurological complications, OR of 0.48 (95% CI 0.26 to 0.86), p=0.015), pulmonary complications, OR: 0.38 (95%CI 0.21 to 0.67), p=0.001) and cardiac complications, OR: 0.24 (95% CI 0.15 to 0.37), p<0.001). The Gore TAG study[19–24] reported endoleak rates for ESG patients of 8.5%.

## DISCUSSION

This is the first systematic review that reports results from non-randomised comparative studies regarding the effectiveness of elective ESG and OSR for arch and or descending TAA. Previous reviews have either included single-arm studies only[30] or were comparative reviews that included non-elective procedures,[9 31] or mixed pathologies such as dissections.[10 32] The results of these reviews may not be applicable to patients undergoing elective procedures for descending or arch TAA pathology.

In this review, only five comparative cohort studies were identified. The lack of RCTs comparing ESG and OSR for elective arch/DTA aneurysms may reflect the lack of equipoise from practitioners and patients. Well-designed non-randomised trials can demonstrate a high degree of concordance with the results of well-designed RCTs,

provided that all confounders are adequately accommodated in the design and/or analysis.[33] However, the studies identified in this review had moderate to serious risk of bias, mainly due to potential for confounding.

Although, complications in the short-term also appear to favour ESG, evidence regarding short-term complications is either lacking or inconsistently reported. All five studies[19 25–28] suggested that the less invasive nature of ESG confers lower short-term mortality and fewer short-term non-fatal complications than OSR. The two small unadjusted studies[19–24 26] had a much larger effect than larger studies.[25 27 28] These larger studies used more robust methods for addressing confounding; this may indicate the presence of publication bias where only those smaller studies with larger effect sizes have been published.

Longer-term results are difficult to interpret since most data were not presented in a way to separate early risk from longer term survival. The two smaller studies[19–24 26] reported continuing survival benefit for ESG patients in the long term but are limited by small numbers, different recruitment periods, and failure to exclude early risk. The survival advantage associated with ESG in the Gore TAG[19–24] study disappears by year 5, which suggests that the hazards are lower for OSR patients that survive the initial procedure. Conversely, the two larger studies (von

Allmen et al[27] and Goodney et al[25] report that the short-term survival advantage of ESG were reversed by twelve months, and that the OSR group had an increasing survival advantage thereafter. Similarly, time to reintervention favoured ESG in the smaller Gore TAG[19–24] study, while time to aortic-related reinterventions favoured OSR in the larger study von Allmen et al.[27]

The evidence regarding mortality is inconsistent across the studies with more recent, larger studies[25 27 28] reporting a more modest survival benefit for ESG in the short term and a survival benefit for OSR in the long-term. The two smaller studies[19–24 26] suggest a large reduction in the risk of mortality and short term complications in the short run and a survival benefit in the long-run for ESG.

Inconsistency of results across studies may be due to heterogeneity of the study populations, both within and across studies. Differences in recruitment periods may reflect changes in technology and practice in both groups. The larger studies[25 27 28] were based on patients recruited up to twenty years later than the smaller studies. Outcomes may have improved differentially with OSR patients receiving proportionally better outcomes due to general improvements in surgical practice and patient selection over time.[34] A potential reason for the reported long-term survival benefit of OSR in more recent, larger studies may be differences in patient risk in each intervention group. OSR is advocated for younger, lower-risk patients and ESG is preferred for older, higher-risk patients.[35] This selection bias may not have been adequately accommodated in adjusted analyses.

The three large studies[25 27 28] were based on administrative health records with case ascertainment that depends on accurate coding, which can vary across and within providers.[36] Furthermore, one of the small studies, Piffaretti et al,[26] recruited patients from one centre which may have compromised external validity as they may, for example, depend on the expertise within a centre and the nature of the patients that centre treats.[37 38] Additionally, the Gore TAG study[19–24] used a more restricted inclusion criteria for ESG patients compared with those who received OSR, potentially introducing heterogeneity between the two groups.

The headline conclusion of this review is that while there is a degree of variability, OSR is better for long-term survival and avoidance of reintervention, and that ESG is better for early outcomes. This conclusion is based on non-randomised comparisons in which the influence of important confounders for choice of intervention (ESG or OSR) has not been adjusted for. Deeper exploration is required, but randomised data may be difficult to acquire as there are well-established guidelines about which patients are favoured for OSR and which for ESG. Thus, to compare ESG and OSR it is first necessary to establish an understanding of the reasons driving clinical management, for example, by a structured consensus analysis such as a Delphi study. Thereafter, unless there is clinical equipoise about which patients are best served by OSR or ESG, the best quality evidence to establish the relative merits of ESG and OSR for arch/DTA aneurysms is likely to be a systematic, prospective, observational, comparative study, using individual patient data.

## Limitations

The findings in this review are limited by the number and potential for confounding of the five studies[19–28] identified, so that definitive conclusions are not possible. The adjustment for confounders is likely to be only partial and some bias in the estimates of differences in mortality and complications may persist, despite attempts to reduce bias in the larger studies identified. Statistical analysis of non-fatal complications was not possible due to limited and inconsistent reporting.

## CONCLUSIONS

This systematic review is the first to consider evidence from non-randomised studies that directly compare ESG and OSR for the treatment of elective arch/DTA aneurysms. Other reviews conducted in this area have been complicated by the inclusion of treatments for emergency cases or other thoracic aortic pathologies (eg, thoracic aortic ulcer, or dissection).

There is limited and increasingly dated evidence addressing the comparison of ESG and OSR in the management of elective arch and or DTA aneurysms. Further, high quality, evidence is required. Although large RCTs may not be possible in this relatively rare condition, future studies should use strong methodological study design in order to control for potential confounders and reduce potential biases. Additionally, given the current conflicting evidence, it is important that comparisons of OSR and ESG are conducted in both the short term and the long term.

**Author affiliations**
[1]Health and Life Sciences, Northumbria University, Newcastle upon Tyne, UK
[2]Cardiothoracic Surgery, John Radcliffe Hospital, Oxford, UK
[3]Medical Statistics, London School of Hygiene and Tropical Medicine, London, UK
[4]Health Economics Group, Population Health Sciences Institute, Newcastle University, Newcastle upon Tyne, UK
[5]Wessex Institute, University of Southampton, Southampton, UK
[6]Southampton Clinical Trials Unit, University of Southampton, Southampton, UK
[7]University Hospital Southampton NHS Foundation Trust, Southampton, UK
[8]Papworth Trials Unit Collaboration, Papworth Hospital NHS Foundation Trust, Cambridge, UK
[9]Cardiac Surgery, Papworth Hospital NHS Foundation Trust, Cambridge, UK

**Acknowledgements** The authors would like to thank our Funders and the ETTAA Working Group for their support with this study.

**Contributors** Author contributions according to the CrediT (contributor roles taxonomy). Conceptualisation: AM, JG, PS, LS, LV, AC, PM and SL; Data curation: AM ;Formal Analysis: AM, JG, PM; Funding acquisition: JG, AC, LV, CF, PS, LS, SL; Investigation: AM, JG, PM, LV, PS; Methodology: AM, JG, PM, LV, AC, CF, PS, LS, SL; Supervision: JG, PM, LV; Validation: AM, JG, PM, PS, LS, PC, SL; Writing–original draft: AM, JG, PM; Writing–review and editing: AM, JG, PM, AC, LV, CF, PS, LS, PC and SL.

**Funding** This report is independent research funded by the National Institute for Health Research (Health Technology Assessment, 11/147/03 - Effective Treatments for Thoracic Aortic Aneurysms (ETT AA study): A prospective cohort study).

**Disclaimer** The views expressed in this publication are those of the author(s) and not necessarily those of the NHS, the National Institute for Health Research or the Department of Health."

**Competing interests** LV reports grants from NIHR Health Technology Assessment Programme, during the conduct of the study; and being a member of the NIHR Health Technology Assessment Programme Clinical Trials and Evaluation Panel from 2015 to 2018. CF reports grants from DoH NIHR HTA Programme, during the conduct of the study. PC reports grants from NIHR, outside the submitted work. SL reports grants from NIHR, during the conduct of the study.

**Patient consent for publication** Not required.

**Provenance and peer review** Not commissioned; externally peer reviewed.

**Data availability statement** All data relevant to the study are included in the article or uploaded as online supplemental information. Data were extracted from the studies included in this systematic review and referenced below.

**ORCID iDs**
Andrew McCarthy http://orcid.org/0000-0002-3385-6302
Andrew Cook http://orcid.org/0000-0002-6680-439X
Carol Freeman http://orcid.org/0000-0001-9751-0784

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
