## [Reviewer comments · BMJ Open]

ARTICLE DETAILS

TITLE (PROVISIONAL)	A Systematic Review of Endovascular Stent Grafting Versus Open Surgical Repair for the Elective Treatment of Arch/Descending Thoracic Aortic Aneurysms
AUTHORS	McCarthy, Andrew; Gray, Joanne; Sastry, Priya; Sharples, Linda; Vale, Luke; Cook, Andrew; Mcmeekin, Peter; Freeman, Carol; Catarino, Pedro; Large, Stephen

VERSION 1 – REVIEW

REVIEWER	Professor Stewart Walsh NUI Galway Ireland
REVIEW RETURNED	05-Nov-2020

GENERAL COMMENTS	The authors have submitted a systematic review they conducted to evaluate mortality and reinterventions following elective endovascular aortic arch or descending aortic repair versus open surgery. The authors have made a valiant effort to provide an objective evaluation in an area dominated by case series and cohort studies. There are several areas that could be addressed to improve the review. Major Comments Results The included studies section is confusing – 5 studies mentioned in text page 9, line 54 onwards. The report then mentions two with non-overlapping or partially overlapping study periods (Gore Tag and Piffaretti). The reader is referred to Figure 1 which concludes with 10 studies (not 5) and table 1 (which lists 7 studies). I appreciate the Gore Tag study generated multiple publications and is treated correctly as a single study – that needs to be represented on the PRISMA flow sheet. A similar approach could be adopted with the two Goodney studies mentioned on Table 1 – given the study periods overlap, is it not highly likely that the two papers listed in Table 1 contain many of the same patients? It is a subjective assessment as to which one to include but personally, I would use the data from the later propensity matched paper. Page 17, lines 10 to 17 – I would be inclined to mention only one of the Goodney studies results on mortality. The study cohorts overlap and so mentioning the two studies as separate entities is essentially ‘double counting’ results. The authors mention that it was not possible to meta-analyse all-cause mortality after discharge or beyond 30 days due to varying duration of follow up but in the narrative section, the bulk (4 out of
---

	5) of the studies seem to have reported 5-year survival? Could death within five years be compared? 4 of the 5 studies considered DTAA only. Only the Hughes paper potentially included procedures on the more proximal aorta, which are more likely to have been open and more likely to generate complications. A sensitivity analysis for unadjusted mortality excluding the Hughes paper should be considered. The review title includes arch repair but the vast bulk of the included patients underwent DTAA repair only. Some discussion of this should be included. To my mind, data from TEVAR on a straightforward DTAA requiring one or 2 stents, possibly under local anaesthetic, should not be compared to open arch replacement requiring cardiopulmonary bypass, to infer a benefit for TEVAR. Minor Comments The title should clearly state that the review only considers elective interventions.
--	---

REVIEWER	KAHLBERG A.L. Vita-Salute University School of Medicine Scientific Institute H. San Raffaele, Milan, Italy
REVIEW RETURNED	05-Nov-2020

GENERAL COMMENTS	Dear authors You surely have written a well conducted systematic review and statistical analysis about a very interesting topic about vascular surgery, trying to find out clear evidence that could guide daily clinical practice. Unfortunately, randomized controlled trials comparing TEVAR and OSR in thoracic aortic aneurysms are still absent; and without this type of literature, it is not possible to conduct an actual analysis of the superiority of one procedure over another. However, I believe that it is also impossible to carry out an RCT on two types of treatment which, nowadays, present bias both in the selection of patients and in the ability to propose a possible type of treatment. Available reports from non-randomized studies suggest that TEVAR is technically feasible (depending on the anatomy) and may reduce the early negative outcomes related to OSR; in spite of that, endovascular surgery presents many late complications uncommon in open surgery, requiring more frequent surveillance without a significant benefit concerning late survival. Nevertheless, is commendable the effort you made in trying to find works as similar as possible and conducting a precise statistical analysis on the few truly comparable elements. Finally, I have few questions about this work. 1- Does this manuscript provide any new data or insights were presented when compared to the existing body of literature on the topic of comparison between TEVAR and OSR in the treatment of DTA? 2- No information is given about the diagnostic process of the patients. Do the authors have any information regarding the pre-operative evaluations to assess patient global clinical conditions?
---

	3- In table 1 appear the citation of “Andrassy et al.” but this study isn’t mentioned into the references or into table 2 and appendix. Why? 4- No information is given about patients suffering for Marfan syndrome or other connective tissue disorders. Do the authors have any information regarding this subgroup of population? How have these patients been treated? 5- The treatment of aortic arch aneurysms usually requires the debranching of the supra-aortic trunks, both for open and endovascular surgery, thus leading to very different and technically more complex interventions than the ones of a descending TAA. Do the authors can better explain which type of aortic arch aneurysms are included into this review? 6- Considering the long-terms outcomes, do the authors have any information about the patients’ quality of life? Do you think it could be a useful item to be evaluated during the follow-up? 7- Can the authors better explain what they mean by aortic-related mortality? 8- Piffaretti et al report OSR up to 2000 and later TEVAR: a certain change of strategy appears evident in the treatment of the descending TAA. Don't you think that this attitude could be a big confounding factor in the analysis you have conducted, such as to invalidate it? 9- Both open and endovascular surgery have changed significantly over the years, with the innovation of materials for TEVAR and with new adjuncts for open surgery. Are the authors able to provide comparative data between the various eras, both as regards the open and endovascular treatment?
--	--

VERSION 1 – AUTHOR RESPONSE

Reviewer: 1
Professor Stewart Walsh
NUI Galway
Ireland

Please state any competing interests or state ‘None declared’:
none declared

Comments to the Author

The authors have submitted a systematic review they conducted to evaluate mortality and reinterventions following elective endovascular aortic arch or descending aortic repair versus open surgery. The authors have made a valiant effort to provide an objective evaluation in an area dominated by case series and cohort studies. There are several areas that could be addressed to improve the review.

Major Comments

Results

The included studies section is confusing – 5 studies mentioned in text page 9, line 54 onwards. The report then mentions two with non-overlapping or partially overlapping study periods (Gore Tag and Piffaretti). The reader is referred to Figure 1 which concludes with 10 studies (not 5) and table 1 (which lists 7 studies). I appreciate the Gore Tag study generated multiple publications and is treated correctly as a single study – that needs to be represented on the PRISMA flow sheet. A similar approach

could be adopted with the two Goodney studies mentioned on Table 1 – given the study periods overlap, is it not highly likely that the two papers listed in Table 1 contain many of the same patients? It is a subjective assessment as to which one to include but personally, I would use the data from the later propensity matched paper.

We agree that the PRISMA diagram was not very clear in relation to the multiple publications of the GoreTAG study. We have changed the PRISMA diagram to show 10 papers meeting the inclusion criteria of which 6 papers belonged to the GoreTAG study resulting in 5 studies being included overall.

The Goodney study consists of a single paper which reports two sets of analyses, reporting mortality for an unadjusted cohort and for a propensity matched cohort that includes a subset of the unadjusted cohort. Our view is that it is important to report results of both, since unadjusted comparisons are consistent with other studies, whilst the propensity-matched analysis is less biased but relates to a different cohort. We agree that having two columns for the Goodney results in table 1 makes it unclear that there is only one paper reporting two sets of analyses. We have therefore combined the two columns together to report both sets of results in one column. Adjusted are tables 1, appendix 2: data extraction table, Appendix 3: Risk of Bias assessment.

Page 17, lines 10 to 17 – I would be inclined to mention only one of the Goodney studies results on mortality. The study cohorts overlap and so mentioning the two studies as separate entities is essentially 'double counting' results.

As mentioned above the propensity cohort is a subgroup of the unmatched cohort of the Goodney paper. We report both sets of results but have clarified when we report the unmatched cohort and when we report the propensity matched cohort. Only one analysis from Goodney is included in each meta-analysis, and to be consistent with other studies, in the unadjusted short-term mortality meta-analysis (fig 2) we use the unmatched Goodney analysis and in the adjusted short-term all-cause mortality meta-analysis (fig 3) we use the propensity matched analysis.

The authors mention that it was not possible to meta-analyse all-cause mortality after discharge or beyond 30 days due to varying duration of follow up but in the narrative section, the bulk (4 out of 5) of the studies seem to have reported 5-year survival? Could death within five years be compared? Unfortunately, there were several methodological differences between studies in their analyses of longer-term mortality, which precluded meta-analysis. As well as different follow-up times studies used different estimates of treatment comparisons. For example, Von Allmen reported OR for 1-year survival and a hazard ratio for 5-year survival, whilst the other three studies that reported 5-year survival provided Kaplan Meier curves, of which the Goodney study adjusts for age sex and other variables whilst GoreTag and Piffaretti do not report controlling for any variables. Thus, we have provided a qualitative comparison of reported 5-year survival. In order to conduct meta-analysis, we would need to access the individual patient data from each study, or at least common summaries of the treatment comparison.

4 of the 5 studies considered DTAA only. Only the Hughes paper potentially included procedures on the more proximal aorta, which are more likely to have been open and more likely to generate complications. A sensitivity analysis for unadjusted mortality excluding the Hughes paper should be considered.

The review title includes arch repair but the vast bulk of the included patients underwent DTAA repair only. Some discussion of this should be included. To my mind, data from TEVAR on a straightforward DTAA requiring one or 2 stents, possibly under local anaesthetic, should not be compared to open arch replacement requiring cardiopulmonary bypass, to infer a benefit for TEVAR.

As suggested we have added a sensitivity analysis excluding the Hughes paper. The treatment comparisons do not differ substantially from the original meta-analysis of unadjusted short-

term mortality, but the estimates had wider 95% CI. (results of all five studies: pooled OR: OR: 0.75; 95 %CI. (0.55-1.03) results of excluding Hughes et al pooled OR: 0.73 (95%CI 0.45, 1.18)

Minor Comments

The title should clearly state that the review only considers elective interventions.

We agree and have changed the title as you suggest

Reviewer: 2

KAHLBERG A.L.

Vita-Salute University School of Medicine

Scientific Institute H. San Raffaele, Milan, Italy

Please state any competing interests or state 'None declared':

None declared

Comments to the Author

Dear authors

You surely have written a well conducted systematic review and statistical analysis about a very interesting topic about vascular surgery, trying to find out clear evidence that could guide daily clinical practice.

Unfortunately, randomized controlled trials comparing TEVAR and OSR in thoracic aortic aneurysms are still absent; and without this type of literature, it is not possible to conduct an actual analysis of the superiority of one procedure over another.

However, I believe that it is also impossible to carry out an RCT on two types of treatment which, nowadays, present bias both in the selection of patients and in the ability to propose a possible type of treatment.

Available reports from non-randomized studies suggest that TEVAR is technically feasible (depending on the anatomy) and may reduce the early negative outcomes related to OSR; in spite of that, endovascular surgery presents many late complications uncommon in open surgery, requiring more frequent surveillance without a significant benefit concerning late survival.

Nevertheless, is commendable the effort you made in trying to find works as similar as possible and conducting a precise statistical analysis on the few truly comparable elements.

Finally, I have few questions about this work.

1- Does this manuscript provide any new data or insights were presented when compared to the existing body of literature on the topic of comparison between TEVAR and OSR in the treatment of DTA?

This review is the first we are aware of to consider evidence of non-randomised studies that directly compare ESG and OSR treatment for arch/descending thoracic aortic aneurysms only. There are many reviews that consider evidence which include other pathologies such as dissections, ruptures or abdominal aneurysm etc. We believe that these other pathologies have their own specific risks with regards to ESG and OSR and therefore including them in a comparison of treatments for elective arch/descending thoracic aortic aneurysms may introduce bias.

2- No information is given about the diagnostic process of the patients. Do the authors have any information regarding the pre-operative evaluations to assess patient global clinical conditions? The studies did not provide information on the preoperative evaluations used to assess global clinical conditions

3- In table 1 appear the citation of "Andrassy et al." but this study isn't mentioned into the references or into table 2 and appendix. Why?

The Andrassy study is a small study that was initially included in the review but failed one of the eligibility criteria (study design criterion) and was subsequently excluded. It's inclusion in table 1 is an editing error and we have removed it.

4- No information is given about patients suffering for Marfan syndrome or other connective tissue disorders. Do the authors have any information regarding this subgroup of population? How have these patients been treated?

No information was provided by any of the studies included in this review about whether patients had connective tissue disorders. Given three of the studies used registry data it is possible that some patients with connective tissue disorders were included. However it is not possible for us to extract this subgroup.

5- The treatment of aortic arch aneurysms usually requires the debranching of the supra-aortic trunks, both for open and endovascular surgery, thus leading to very different and technically more complex interventions than the ones of a descending TAA. Do the authors can better explain which type of aortic arch aneurysms are included into this review?

Only Hughes et al reported thoracic arch descending aneurysms. We have added a sensitivity analysis removing this study to address any potential bias from the inclusion of arch aneurysms. The results of this sensitivity analysis did not change the overall comparisons of unadjusted mortality (the only outcome reported by Hughes et al 2014).

6- Considering the long-terms outcomes, do the authors have any information about the patients' quality of life? Do you think it could be a useful item to be evaluated during the follow-up?

We agree that it would be useful to evaluate quality of life when comparing OSR and ESG. However, no studies provided quality of life data and so we could not investigate this.

7- Can the authors better explain what they mean by aortic-related mortality?

Aortic-related mortality was as defined in the studies that reported it. We have added the definitions (below) to the manuscript in the results section.

Van Allmen defined aortic-related mortality as any aortic-related death occurring within 5 years of the index operation according to ICD-10 codes referring to aorta or cause of deaths closely related to complications of surgery (e.g. paraplegia)

GoreTAG defined aneurysm-related mortality as those that occurred in hospital or within 30 days of initial procedure or any reintervention or a death due to the aneurysm or the treatment device.

8- Piffaretti et al report OSR up to 2000 and later TEVAR: a certain change of strategy appears evident in the treatment of the descending TAA. Don't you think that this attitude could be a big confounding factor in the analysis you have conducted, such as to invalidate it?

The Piffaretti study is a very small single centre study, with non-overlapping time periods for the two interventions, which does cast doubt on the comparability of the two groups. However, since the historical OSR cohort is matched with the ESG cohort on demographic data and risk factors, it satisfies the eligibility criteria of the review. In addition, as can be seen in the meta-analysis of unadjusted mortality in figure 2, the Piffaretti study is weighted to contribute only 1.5% towards the pooled OR, so that it has very little impact on the meta-analysis result.

9- Both open and endovascular surgery have changed significantly over the years, with the innovation of materials for TEVAR and with new adjuncts for open surgery. Are the authors able to provide comparative data between the various eras, both as regards the open and endovascular treatment?

Four of the studies included in this review consist of data prior to 2008. Only one study (Von Allmen) includes data from 2006-2011 and this has similar results to the other two large registry studies (Hughes et al and Goodney et al). There is insufficient information provided by the studies to conduct a comparison between eras.

VERSION 2 – REVIEW

REVIEWER	Stewart Walsh NUI Galway, Ireland
REVIEW RETURNED	19-Jan-2021

GENERAL COMMENTS	Thank you for asking me to review this revision. Overall, the manuscript has significantly improved. I only have a couple of comments. The situation regarding the two Goodney papers is much clearer now. I would consider replacing the sentence ‘This review reports information from both cohorts separately but does not include both in any analysis to avoid double-counting’ in results with something like ‘Data from the unadjusted cohort were used in all meta-analyses except the adjusted mortality meta-analysis, which used the propensity-matched data’. Thank you for including the sensitivity analysis. I am not sure it is accurate to state that ‘Overall, the odds of short-term mortality were lower in the ESG group than the OSR group (OR: 0.73; 95 %CI. (0.45-1.18))’ at the start of the second paragraph. The 95%CI is well across the line of no effect and the associated p-value in Figure 3 is 0.2. That said, the point estimates of all four studies do favour ESG. Consider replacing the sentence with ‘While the point effect estimates all favoured ESG, the overall pooled odds ratio of 0.73 (95%CI 0.45 to 1.18) did not reach statistical significance (p=0.2). A line regarding the sensitivity analysis should be added to the abstract. Minor Comments Abstract: PROSPERO not POSPERO
--

REVIEWER	Andrea Kahlberg Vita-Salute University School of Medicine Scientific Institute H. San Raffaele, Milan, Italy
REVIEW RETURNED	29-Jan-2021

GENERAL COMMENTS	The authors responded satisfactorily to the reviewer's requests. Despite some limitations connected with the absence of clear data in the literature, I believe that the manuscript is well written and worthy of publication.
--

VERSION 2 – AUTHOR RESPONSE

1. The situation regarding the two Goodney papers is much clearer now. I would consider replacing the sentence ‘This review reports information from both cohorts separately but does not include both in any analysis to avoid double-counting’ in results with something like ‘Data from the unadjusted cohort were used in all meta-analyses except the adjusted mortality meta-analysis, which used the propensity-matched data’.

We agree with this suggestion and have changed the sentence accordingly.

2. Thank you for including the sensitivity analysis. I am not sure it is accurate to state that 'Overall, the odds of short-term mortality were lower in the ESG group than the OSR group (OR: 0.73; 95 %CI. (0.45-1.18))' at the start of the second paragraph. The 95%CI is well across the line of no effect and the associated p-value in Figure 3 is 0.2. That said, the point estimates of all four studies do favour ESG. Consider replacing the sentence with 'While the point effect estimates all favoured ESG, the overall pooled odds ratio of 0.73 (95%CI 0.45 to 1.18) did not reach statistical significance (p=0.2).

We agree with this and have changed the sentence accordingly

3. A line regarding the sensitivity analysis should be added to the abstract.

We agree with this and had added the results of the sensitivity analysis to the abstract

4. Minor Comments

Abstract: PROSPERO not POSPERO

Corrected to PROSPERO in the abstract